


# Systematic global evaluation of accuracy of seasonal climate forecasts for monthly precipitation of JMA/MRI-CPS2 by comparing with a statistical system using climate indices

Yuji Masutomi[1, 2], Toshichika Iizumi[3], Kei Oyoshi[4], Nobuyuki Kayaba[5,6], Wonsik Kim[3], Takahiro Takimoto[3], Yoshimitsu Masaki[2]

[1]Center for Climate Change Adaptation, National Institute for Environmental Studies, 16-2 Onogawa, Tsukuba, Ibaraki 305-8506, Japan
[2]College of Agriculture, Ibaraki University, 3-21-1 Chuo, Ami, Inashiki, Ibaraki 300-0393, Japan
[3]Institute for Agro-Environmental Sciences, National Agriculture and Food Research Organization, 3-1-3 Kannondai, Tsukuba, Ibaraki 305-8604, Japan
[4]Earth Observation Research Center, Japan Aerospace Exploration Agency, 2-1-1 Sengen, Tsukuba, Ibaraki 305-8505, Japan
[5]Japan Meteorological Agency, 3-6-9 Toranomon, Minato City, Tokyo 105-8431, Japan
[6]Meteorological Research Institute, 1-1 Nagamine, Tsukuba, Ibaraki 305-0052, Japan

*Correspondence to*: Yuji Masutomi (masutomi.yuji@nies.go.jp)

**Abstract.** In this study, we aimed to evaluate the monthly precipitation forecasts of JMA/MRI-CPS2, a global dynamical seasonal climate forecast (Dyn-SCF) system operated in the Japan Meteorological Agency, by comparing them with the forecasts of a statistical SCF (St-SCF) system using climate indices systematically and globally. Accordingly, we developed a new global St-SCF system using 18 climate indices and compared the monthly precipitation of this system with those of JMA/MRI-CPS2. Consequently, it was found that JMA/MRI-CPS2 forecasts are superior to St-SCFs around the equator (10º S–10º N) even for six-month lead forecasts. For one-month lead forecasts, the accuracy of JMA/MRI-CPS2 forecasts was higher than that of St-SCFs when viewed globally. In contrast, for forecasts made two months or longer in advance, St-SCFs had an advantage in global forecasts. In addition to evaluating the accuracy of JMA/MRI-CPS2 forecasts, the slow dynamics of the ocean and atmosphere, not reproduced by the JMA/MRI-CPS2 system, were determined by comparing the evaluations, and it was concluded that this could contribute to improving Dyn-SCF systems.

## 1 Introduction

Seasonal climate forecasts (SCFs), which predict the weather more than two weeks to one year in advance, can provide useful information for decision-making and early warning in various fields, such as agriculture and water resource management (Doblas-Reyes et al., 2006; Jones et al., 2000; Klemm and McPherson, 2017; Meinke and Stone, 2005; Pozzi et al., 2013); however, their usefulness substantially depends on the accuracy of forecasts. Therefore, the evaluation of the accuracy of SCFs is an important aspect in the construction of SCF systems (Kim et al., 2012), which is implemented in most SCF systems.





The general approach for evaluating the accuracy of SCFs involves analyzing the degree of similarity with the observed data. As a more advanced approach, the assessment of added values compared to the SCF system using climatology or simple statistical methods has been proposed (Luo et al. 2012; Pappenberger et al. 2015; Turco et al. 2017). For dynamical SCF

(Dyn-SCF) systems, in particular, which use dynamical climate models with a large computational load, such as coupled atmosphere-ocean models, it is necessary to show added values as benefits to the large cost by comparing with the forecast accuracy of simple methods with a smaller cost. The mean square skill score is often used to evaluate the forecast accuracy of SCF systems (Stockdale et al. 2011; Kim et al. 2016), and it can be considered an evaluation of added values to SCFs using climatology.

Statistical SCF (St-SCF) systems are an alternative and simpler method for Dyn-SCF systems (Doblas-Reyes et al., 2013). The forecast accuracy of St-SCF and Dyn-SCF systems has been compared in various manners and regions (Folland et al. 1991; Anderson, van den Dool, and Ploshay 1999; Barnston, Glantz, and He 1999; van Oldenborgh et al. 2005; Quan et al. 2006; Wu et al. 2009; Pappenberger et al. 2015; Turco et al. 2017). However, to the best of our knowledge, no systematic global comparison has yet been made because it is difficult to develop a global system for St-SCFs (Eden et al., 2015).

Systematic global comparisons can be used to identify regions and seasons in which Dyn-SCF systems have advantages and disadvantages in forecasting.

Among the various statistical methods used in St-SCF systems, numerous studies have used climate indices like Nino 3.4, Southern Oscillation Index, Madden-Julian Oscillation (Quayle 1929; Nicholls, McBride, and Ormerod 1982; McBride and Nicholls 1983; Gordon 1986; Chu 1989; Stone, Hammer, and Marcussen 1996; Chiew et al. 1998; Kirono, Chiew, and Kent

2010; Schepen, Wang, and Robertson 2012; Eden et al. 2015; Singh and Qin 2020). The predictability of St-SCFs using climate indices relies on the slow dynamics in the ocean and atmosphere, and the climate states are correlated to the slow dynamics. This is essentially the same for Dyn-SCF systems, whose predictability also depends on the presence of slow variations in soil moisture, snow cover, sea ice, and ocean surface temperature (Doblas-Reyes et al., 2013). Therefore, the forecast accuracy of St-SCFs using climate indices can be a suitable benchmark for that of Dyn-SCFs. In addition, by

comparing Dyn-SCFs and St-SCFs, the dynamics that are insufficiently reproduced in Dyn-SCF systems can be clarified, which could contribute to improving the accuracy of Dyn-SCFs.

A global Dyn-SCF system, JMA/MRI-CPS2 (Takaya et al. 2018), developed by the Japan Meteorological Agency (JMA) and Meteorological Research Institute (MRI), is used for operational seasonal forecasting. Takaya et al. (2018) reported that JMA/MRI-CPS2 generally improved the forecast accuracy of the interannual variability in the ocean and atmosphere,

including El Niño events, compared to its predecessor model, JMA/MRI-CPS1 (Takaya et al., 2017). The Tokyo Climate Center, a regional climate center of the World Meteorological Organization, published the monthly forecast accuracy of JMA/MRI-CPS2. For example, the evaluation showed that the accuracy of precipitation forecasts is high near the equator; moreover, the one-month lead forecasts have the highest accuracy in February and the lowest in April. However, comparisons with St-SCFs have not been performed for JMA/MRI-CPS2 forecasts.



In this study, we attempted to evaluate the forecast accuracy of JMA/MRI-CPS2 for monthly precipitation by comparing it with that of the St-SCF system using climate indices. Correspondingly, the forecast accuracy of JMA/MRI-CPS2 was evaluated by comparing it with the observed precipitation. Next, an St-SCF system using 18 climate indices was newly developed and it was used to make monthly precipitation forecasts. Then, the forecast accuracy of the St-SCF system using climate indices was evaluated and compared to that of JMA/MRI-CPS2. In addition, we discussed the possibility of model improvement through comparing Dyn-SCFs with St-SCFs using climate indices.

## 2 Method, model, and data

### 2.1 Method

The outline of this study is presented in Figure 1 and Table 1. In Step 1, the accuracy of monthly precipitation forecasts of JMA/MRI-CPS2 was evaluated through comparison with observed data. The hindcast data of JMA/MRI-CPS2 from 1981 to 2010 were used for monthly precipitation forecasts. Monthly precipitation data from the Global Precipitation Climatology Project (GPCP (Adler et al., 2003, 2018)) v2.3, provided by NOAA/OAR/ESRL PSL, were used as observations. The study area was global and the spatial resolution for the comparison was 2.5° × 2.5°; moreover, the accuracy was evaluated for each grid. The spatial resolution of both JMA/MRI-CPS2 and GPCP v2.3 was 2.5° × 2.5°; however, the center of grids was not matched between them. Thus, GPCP v2.3 was re-gridded to the grid of JMR/MRI-CPS2 using the bilinear method. The hindcast data of JMA/MRI-CPS2 included five ensembles with different initial conditions, and the ensemble mean was used in this study.

Next, in Step 2-1, statistical models were constructed for each grid and 18 climate indices, with monthly values of a climate index as an explanatory variable and monthly precipitation as an objective variable (see Section 2.2.2). The smoothing spline method was used to create the statistical models. In Step 2-2, monthly precipitation for each year was forecasted in each grid using the leave-one-out method. In Step 2-3, the forecast values of 18 statistical models were compared with GPCP v2.3, and the forecast value of the model with the highest correlation with observations was selected for each grid to create a composite forecast value of statistical models. Simultaneously, a composite forecast accuracy was obtained by combining the forecast accuracy of selected forecast values in each grid.

Finally, in Step 3, the forecast accuracy of JMA/MRI-CPS2 forecasts, obtained in Step 1, was compared with the composite forecast accuracy of the statistical models obtained in Steps 2-3.

The anomaly correlation coefficient (ACC) between the forecast and observed values was used to evaluate the forecast accuracy, and the deviation in climatology from 1981 to 2010 for the forecasts and observations was used to calculate the ACC. A significance level of 0.05 was used to evaluate statistical significance of ACC. Forecasts with 1 to 6 lead months were evaluated. For example, in the case of precipitation forecasts for October, monthly precipitation forecasts that started from April to September were used. In the statistical model for forecasts in October, the precipitation in October was used as an objective variable and climate indices from April to September were used as explanatory variables.



## 2.2 Model and data

### 2.2.1 JMA/MRI-CPS2

The main component of JMA/MRI-CPS2 is a coupled atmosphere-ocean model (JMA/MRI-CGCM2), whose atmospheric
component is based on the low-resolution version of the JMA Global Spectral Model (GMS1011C, Japan Meteorological
Agency, 2013). Its spatial resolution is TL159 (approximately 110 km) with 60 vertical layers. The ocean component of
JMA/MRI-CGCM2 is based on the MRI Community Ocean Model version 3 (MRI.COM3 v3 (Tsujino, 2010)), which
includes a sea ice model. The spatial resolution of MRI.COPM3 v3 is 1° east-west, 0.3–0.5° north-south, and 52 vertical
layers. The Japanese 55-year Reanalysis (JRA-55; (Kobayashi et al. 2015)) was used to initialize the atmospheric data, and
the Global Ocean Data Assimilation System (MOVE/MRI.COM-G2 (Toyoda et al., 2013)) was used for ocean data.

The JMA/MRI-CPS2 hindcast data were obtained from the Japan Meteorological Business Support Center. The hindcast
period was 1979–2019 and the time resolution was daily. In this study, the hindcast data from 1981 to 2010 were used, and
daily values were averaged to produce monthly values. The spatial resolution of the hindcast data was 2.5° × 2.5°. The
hindcast data included five ensembles with different initial conditions that were averaged for each grid. There were two
forecasts starting in the middle and end of each month. The one closer to the end of the month was used in this study, such as
Jan 31, Feb 25, Mar 27, Apr 26, May 31, Jun 30, Jul 30, Aug 29, Sep 28, Oct 28, Nov 27, and Dec 27. The forecast period of
hindcast data was 240 days; the forecast values in the first six months were used in this study.

### 2.2.2 Statistical seasonal climate forecast system using climate indices

Statistical models were constructed with a climate index as the explanatory variable and precipitation as the objective
variable (in Step 2-1 of Fig. 1). The model is expressed as follows:

$$PRE_{i,j,LM}(T) = \max\{f_{i,j,T,LM}(IDX_j(T - LM)),0\}$$

where $PRE_{i,j,LM}(T)$ denotes the forecast values of precipitation for grid $i$, climatic index $j$, lead month $LM$, and forecast
month $T$. $IDX_j(T - LM)$ is the value of climatic index $j$ in $T - LM$. $f_{i,j,T,LM}$ is a function to obtain precipitation in $T$ for grid $i$
from climatic index $j$ in $T - LM$. In this study, the smoothing spline method was used to develop functions $f_{i,j,T,LM}$. An
example of this function is shown in Fig. 2.

Table 4 summarizes the 18 climate indices used in this study by category. These indices were selected from those provided
by the NOAA Physical Sciences Laboratory, and the values were updated within approximately a week after the end of each
month.



## 3 Results

### 3.1 Comparison of global prediction skill

Fig. 3 shows the global averages of monthly ACC for JMA/MRI-CPS2 forecasts for each lead month. It can be observed that JMA/MRI-CPS2 has high accuracy in one-month lead forecasts; the accuracy decreases rapidly in the two-month lead forecasts, and gradually declines thereafter. The highest accuracy of one-month lead forecasts was observed in February, with an ACC of 0.340, while the worst forecast accuracy was observed in April, with an ACC of 0.224. For two-month lead forecasts, February had an ACC of 0.139, which is less than half the value of one-month lead forecasts. Comparing the ACCs of the ocean and land, it is evident that the one-month lead forecast accuracy in February for land is higher than that for ocean, and that in April for land is lower than that for ocean. For forecasts made more than two months in advance, the forecast accuracy is generally higher for the ocean than land.

Fig. 4 shows the global averages of the monthly ACC for St-SCFs using climate indices for each lead month. It can be seen that the forecast accuracy decreases as the lead month increases, but the decrease is significantly smaller than that in the case of JMA/MRI-CPS2. The highest accuracy of one-month lead forecasts for the global forecast was observed in October, with an ACC of 0.313, while the worst forecast accuracy was observed in June, with an ACC of 0.263. The ACCs of October for two-month and six-month lead forecasts were 0.307 and 0.250, respectively. Comparing the ocean and land areas, it can be observed that the forecast accuracy is higher for ocean forecasts from one to six months in advance.

Fig. 5 shows a comparison of the annual mean ACC between JMA/MRI-CPS2 and St-SCFs using climate indices. It is evident that the forecast accuracy of JMA/MRI-CPS2 is lower than that of St-SCFs even after one month in advance. The difference becomes larger for two months in advance and gradually increases for longer forecasts. However, the global averages of ACC include grids where the correlation is not significant. Fig. 5 shows a comparison of the ratio of areas with significant ACC and that with significant and higher ACC between JMA/MRI-CPS2 and St-SCFs using climate indices. It can be observed that these two values are higher in JMA/MRI-CPS2 for one-month lead forecasts. Therefore, it can be concluded that the forecast accuracy of JMA/MRI-CPS2 is generally higher for one-month lead forecasts. However, when the forecasts were longer than two months, the accuracy of St-SCFs using climate indices was higher.

### 3.2 Spatial comparison of global prediction skill

Fig. 6 shows the spatial distribution of ACC for JMA/MRI-CPS2 in February, April, and June. It can be observed that in February, the month with the highest accuracy in one-month lead forecasts, the area with significant ACC is spread worldwide. However, in April, the month with the lowest accuracy, significant areas are limited to low latitudes near the equator even in one-month lead forecasts. For two-month lead forecasts, the significant areas are more limited, even in February.

Fig. 7 shows the spatial distribution of ACC for St-SCFs in April, June, and October. In October, which had the highest accuracy for one-month lead forecasts, the ACC is unevenly distributed and high, especially in Southeast Asia, Middle East,





East Africa, and equatorial Pacific region. In June, the month with the lowest forecast accuracy, there were no regions with high forecast accuracy, except for the low latitudes near the equator in the western longitude. The regions with high forecast accuracy in October remained highly accurate even for more than two-month lead forecasts. This is a considerable difference from the forecasts of JMA/MRI-CPS2.

Fig. 8 shows the annual mean ACC, ratio of areas with significant ACCs, and ratio of areas with significant and higher ACCs by latitude for JMA/MRI-CPS2 (left) and St-SCFs using climate indices (right). As shown in the maps, the accuracy of JMA/MRI-CPS2 forecasts is generally high at low latitudes, and the difference between one- and two-month lead forecasts is large. St-SCFs using climate indices showed that the forecast accuracy was high at low latitudes, as well as JMA/MRI-CPS2. In contrast, unlike JMA/MRI-CPS2, the statistical forecasts exhibited small differences between one-
month lead forecasts and more than two-month lead forecasts, especially above 20º S and 20º N. For JMA/MRI-CPS2 forecasts, the forecast accuracy decreased as the latitude increased, while the accuracy of St-SCFs remained almost the same above 20º S and 20º N.

Fig. 8 (at the bottom) also shows that JMA/MRI-CPS2 has larger areas with higher ACCs around the equator between 10° S and 10° N from one- to six-month lead forecasts. Even at higher latitudes between 40º S and 70º N, JMA/MRI-CPS2 has
larger areas with higher ACCs in one-month lead forecasts. This is almost true for the ratio of areas with significant (at the middle in Fig. 8).

### 3.3 Regional comparison of global prediction skill: Europe in April

The ACCs for April in Figs. 6 and 7 show that St-SCFs using climate indices have significant ACCs in Europe from one to three months in advance, while JMA/MRI-CPS2 forecasts have no or small significant correlations in Europe for one to
three months in advance. Fig. 9 shows the relationship between the NINO3.4 index and precipitation in Paris (2.5° E and 50º N. This shows that there is a nonlinear correlation between NINO3.4 and precipitation, wherein the precipitation increases with positive (=El Niño) and negative (=La Niña) values of NINO3.4, and reaches a minimum of approximately 0 for NINO3.4. In addition, a nonlinear relationship can be observed from one- to six-month lead forecasts. The slow dynamics responsible for the robust relationship between NINO3.4 and precipitation should exist; however, they are not reproduced in
JMA/MRI-CPS2, implying that further analysis and incorporation of these dynamics into JMA/MRI-CPS2 can improve the forecast accuracy of the model.

### 4 Discussion

### 4.1 Prediction skill of JMA/MRI-CPS2 in comparison to St-SCF

The forecast accuracy of JMA/MRI-CPS2 was evaluated by Takaya et al. (2018) and published by the Tokyo Climate Center.
The evaluation showed that the accuracy of precipitation forecasts is high around the equator and that of one-month lead forecasts is highest in February and lowest in April. The same forecast accuracy was confirmed in this study (Figs. 3, 6, and





8). In addition, by comparing with St-SCFs using climate indices as benchmark, we identified the regions and lead periods in which JMA/MRI-CPS2 has advantages and disadvantages. For example, it was found that JMA/MRI-CPS2 has higher accuracy around the equator between 10º S and 10º N even in long-term forecasts of six months (Fig. 8). In general, it is well known that Dyn-SCF systems have particularly high accuracy in the tropics (Doblas-Reyes et al., 2013). In addition, we showed that JMA/MRI-CPS2 has higher accuracy in comparison to an St-SCF system. To the best of our knowledge, this is the first study that demonstrates Dyn-SCF systems have the added value on seasonal climate forecasts in tropics in comparison with St-SCF systems. It was also found that the accuracy of one-month lead forecasts was higher than that of St-SCFs globally (Fig. 5). This is also considered to be a significant added value of JMA/MRI-CPS2. In contrast, for forecasts longer than two months, St-SCFs using climate indices have an advantage globally (Figs. 3, 4, and 5). At low and middle latitudes between 40º S and 70º N, JMA/MRI-CPS2 has an advantage in one-month forecasts; however, St-SCFs have an advantage in two-month and longer forecasts (Figs. 6, 7, and 8). These results clearly indicate that improving the accuracy of JMA/MRI-CPS2 for longer-term forecasts over two months is a challenge that must be addressed. The improvement in accuracy of Dyn-SCFs in comparison to St-SCFs is discussed in the next section.

## 4.2 Improvement of prediction skill by comparing with St-SCF using climate indices

Various methods have been proposed to improve the forecast accuracy of Dyn-SCFs, including the initialization of soil moisture (Prodhomme et al., 2016b) and incremented resolution (Prodhomme et al., 2016a). For JMA/MRI-CPS2, Takaya et al. (2021) showed that the forecast accuracy increases significantly with the number of ensembles. In this study, by comparing its accuracy with that of St-SCFs, we found the presence of robust slow dynamics that are not well-reproduced by JMA/MRI-CPS2, implying the possibility of improving the accuracy of Dyn-SCFs by incorporating the slow dynamics into the Dyn-SCF system. The key to this approach for model improvement is that both Dyn-SCFs and St-SCFs with climate indices rely on slow dynamics in the ocean and atmosphere. Therefore, a comparison between them can clarify the slow dynamics that are not well-reproduced by Dyn-SCFs. It is expected that this approach will be widely applied to improve the forecast accuracy of Dyn-SCFs.

## 4.3 Global St-SCF system

Several studies have compared Dyn-SCFs and St-SCFs, but no systematic global comparison has been performed (Eden et al., 2015). Therefore, to the best of our knowledge, this is the first study in which the global accuracy of Dyn-CSFs is compared with St-SCFs systematically and globally. Systematic global comparisons have not been performed before because it is difficult to construct global St-SCFs. Eden et al. (2015) recently proposed the construction of a global St-SCF system using multiple climate indices. We used this approach to construct a global St-SCF system. In this regard, this study is more advanced than that by Eden et al. (2015) because we used more climate indices and statistical models that considered the nonlinear relationship between climate indices and precipitation. While Eden et al. (2015) used eight climate indices, including $CO_2$ concentration, to construct their global St-SCF system, we used 18 climate indices in this study. In addition,

Eden et al. (2015) used a linear regression model in their system, but we used the smoothing spline method that can consider the nonlinear relationship between climate indices and precipitation. Moreover, as shown in Figs. 2 and 9, there is a nonlinear relationship between climate indices and precipitation.

Global St-SCF systems can be significantly improved. Moreover, a large number of climate indices can be considered. In this study, we selected 18 climate indices whose values were updated within approximately a week after the end of each month from those provided by the NOAA Physical Sciences Laboratory. The number of available climate indices can be increased by ensuring that these indices are updated. In addition, statistical methods other than the smoothing spline method used in this study can also be utilized. Recently, St-SCFs using machine learning, such as artificial intelligence, have been proposed. If the goal is to improve the accuracy of St-SCF system forecasts, these approaches are also likely to be effective.

## 5 Conclusion

It is concluded that, on a global scale, the forecast accuracy of JMA/MRI-CPS2 for monthly precipitation was observed to be generally higher for one-month lead forecasts; however, St-SCFs were more accurate for forecasts more than two months in advance. Spatially, JMA/MRI-CPS2 has an advantage in forecasting 1–6 months in advance around the equator (10º S–10º N) and in one-month lead forecasts at low and middle latitudes (40º S and 70º N). The comparison with St-SCFs using climate indices suggests the possibility of model improvement for Dyn-SCFs. A more detailed analysis of the comparison and practical improvements for the Dyn-SCF system are expected.

## Data and code availability

The JMA/MRI-CPS2 hindcast data can be purchased from the Japan Meteorological Business Support Center (http://www.jmbsc.or.jp/en/index-e.html). GPCP v2.3 is available at https://psl.noaa.gov/data/gridded/data.gpcp.html. The URLs where the 18 climate indices are obtained are listed in Table 4. All of the source codes used for the analyses in the present paper are stored at https://doi.org/10.5281/zenodo.5090304, while the source code of JMA/MRI-CPS2 is not opened for the public.

## Author contribution

The authors designed the experiments, discussed the results, and reviewed the draft. YM performed the analysis and wrote the original draft.





**Acknowledgment**

This study was partly supported by the Coordination Fund for Promoting AeroSpace Utilization (JPJ000959) from the Ministry of Education, Culture, Sports, Science and Technology (MEXT) and KAKENHI (JP19H03069) from the Japan Society for the Promotion of Science (JSPS).

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

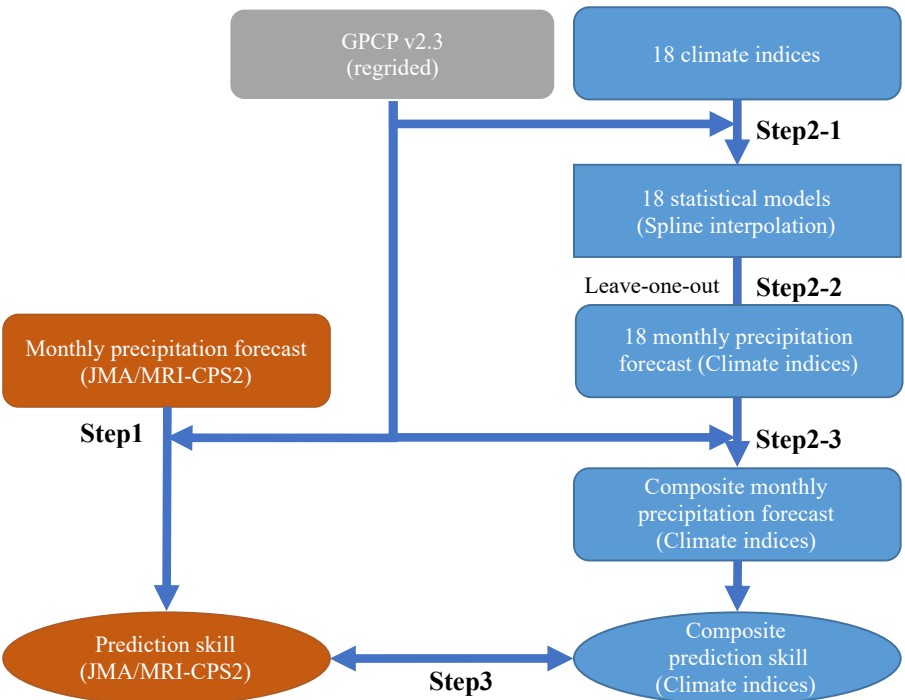

**Figure 1: Research outline**


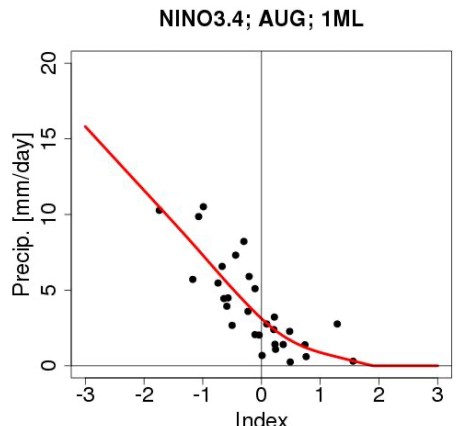

**Figure 2: Spline interpolation curve (red line) of NINO3.4 in July for estimating August precipitation at 110° longitude and -2.5°**

**latitude. Plots denote observational precipitation and the values of NINO3.4 index.**






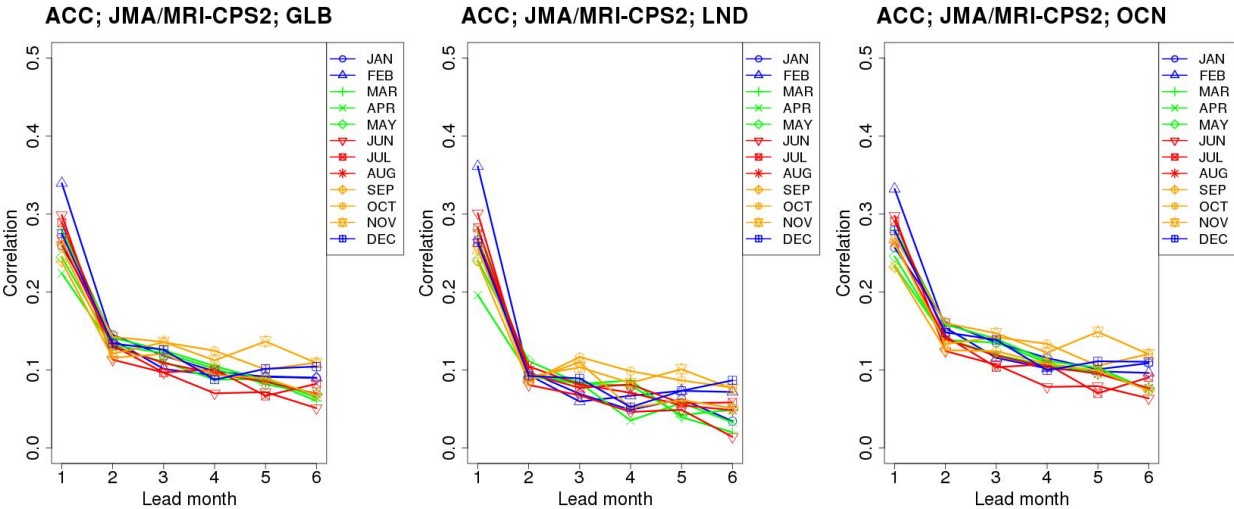

**Figure 3: Globally averaged ACC by JMA/MRI-CPS2 (left: global average (GLB); center: average over land (LND); right: average over ocean (OCN))**


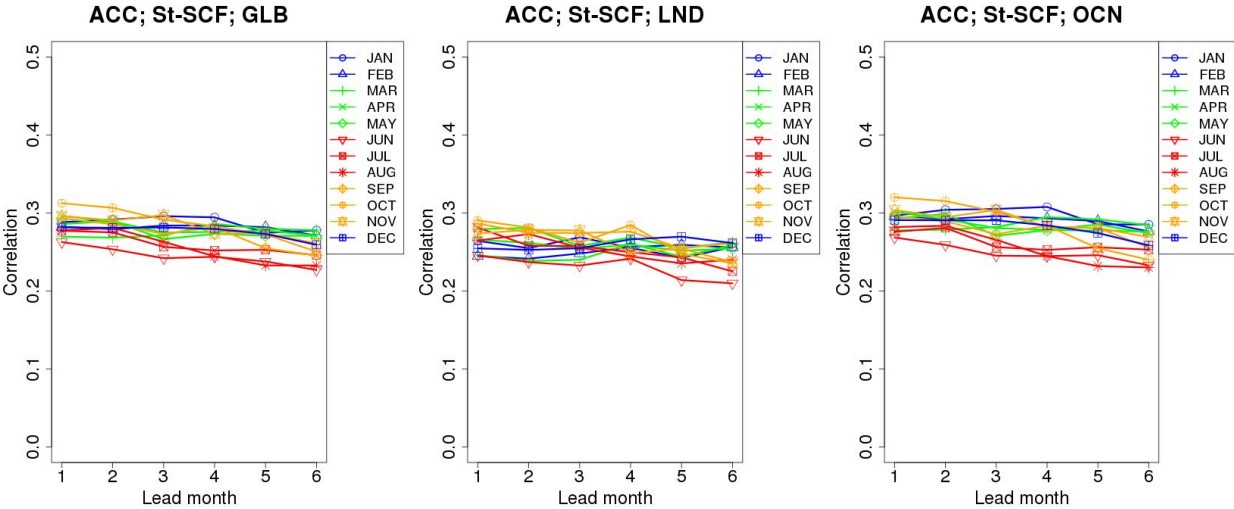

**Figure 4: Globally averaged ACC by St-SCFs (left: global average (GLB); center: average over land (LND); right: average over ocean (OCN))**






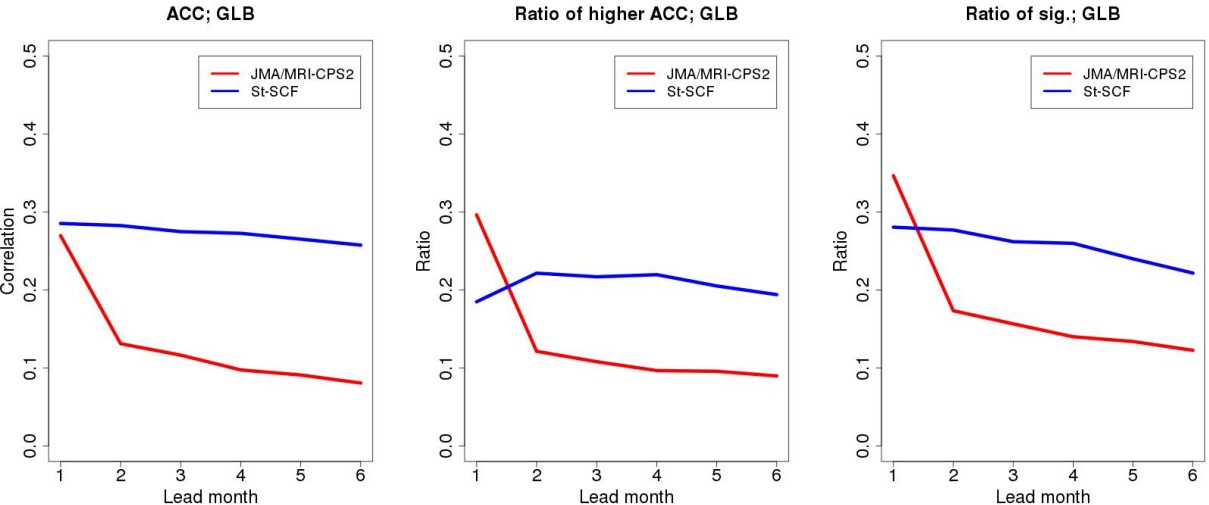

**Figure 5: Comparison of globally averaged annual ACC (left), ratio of area with significant ACC (center), and ratio of higher ACC with significant between JMA/MRI-CPS2 and St-SCFs using climate indices (right)**


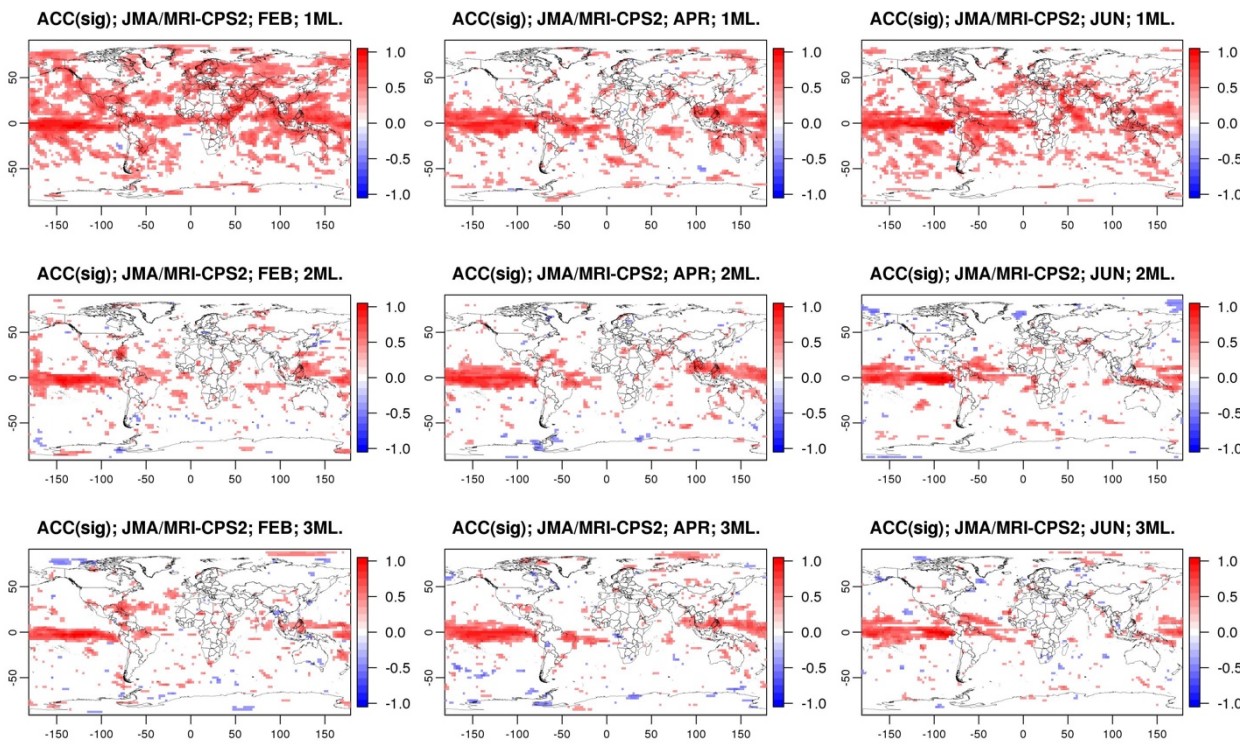

**Figure 6: Spatial distribution of ACC for JMA/MRI-CPS2. Grided with significant are shown. Left, center, and right columns denote Feb, Apr, and Jun, respectively. Top, middle, and bottom denote 1 to 3 months lead.**




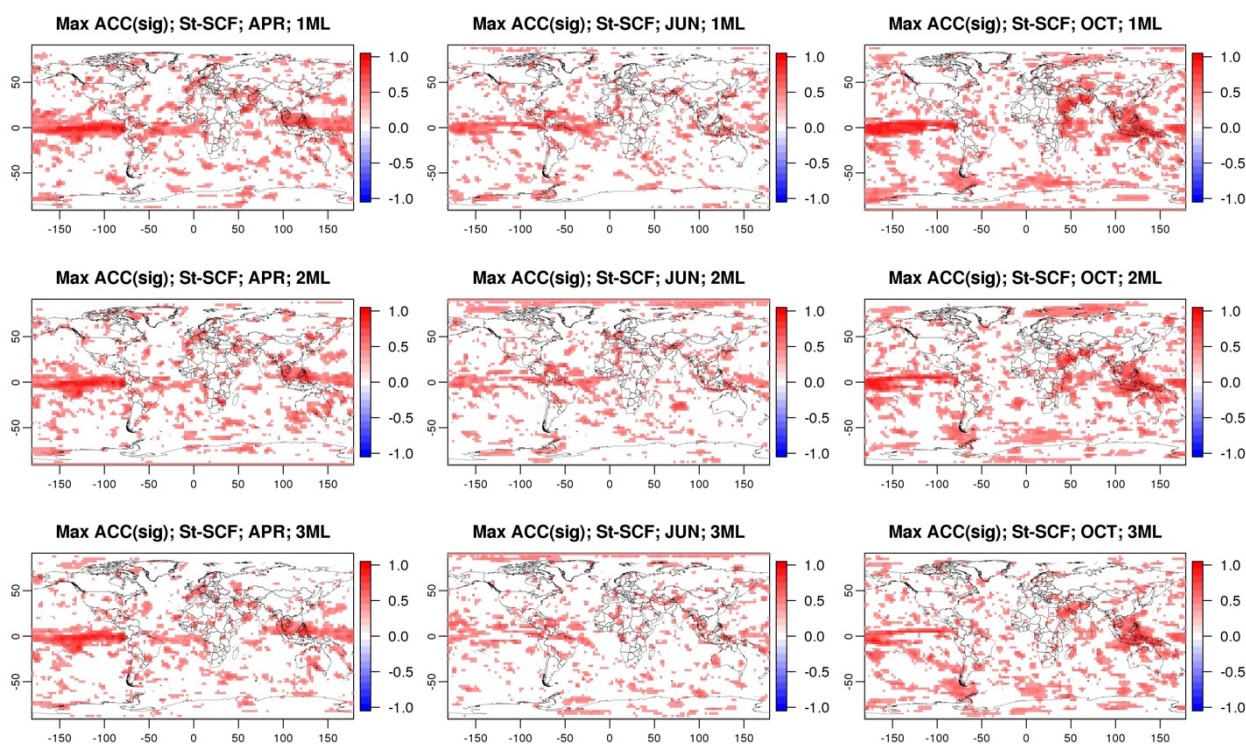


**Figure 7: Spatial distribution of ACC for St-SCFs using climate indices. Grided with significant are shown. Left, center, and right columns denote Apr, Jun, and Oct, respectively. Top, middle, and bottom denote 1 to 3 months lead.**

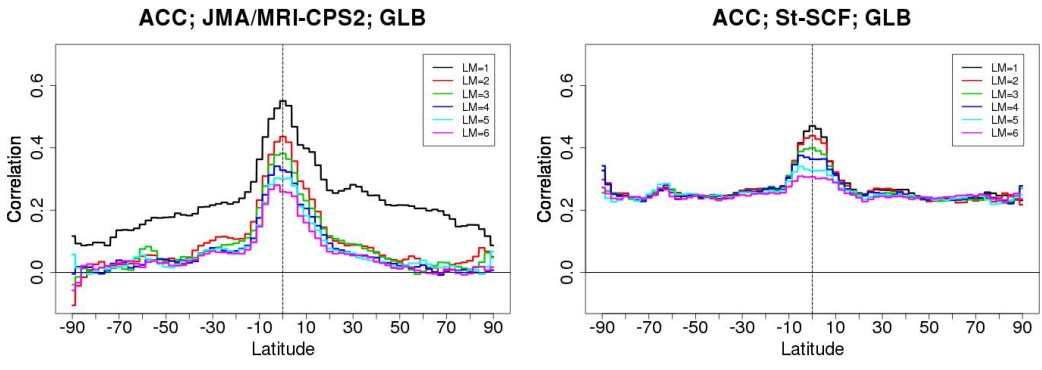




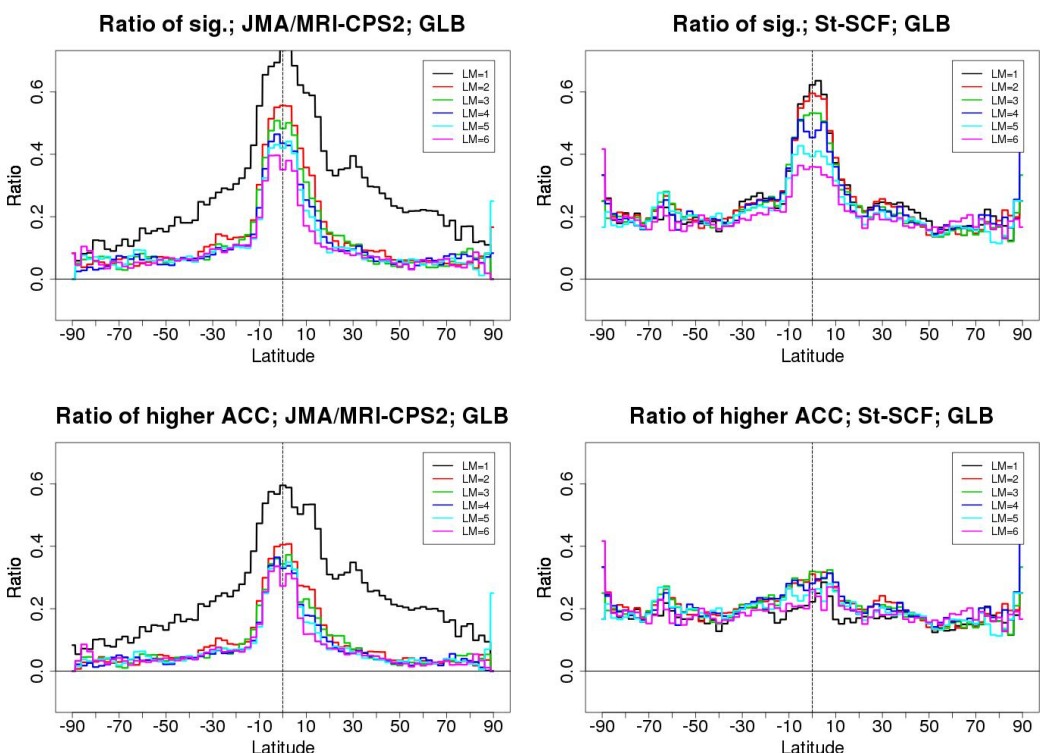

**Figure 8: Comparison of latitude for JMA/MRI-CPS2 (left) and SCFs using climate indices (right). Top: ACC, Middle: ratio of significant ACC; Bottom: ratio of higher ACC between JMA/MRI-CPS2 and St-SCFs using climate indices.**


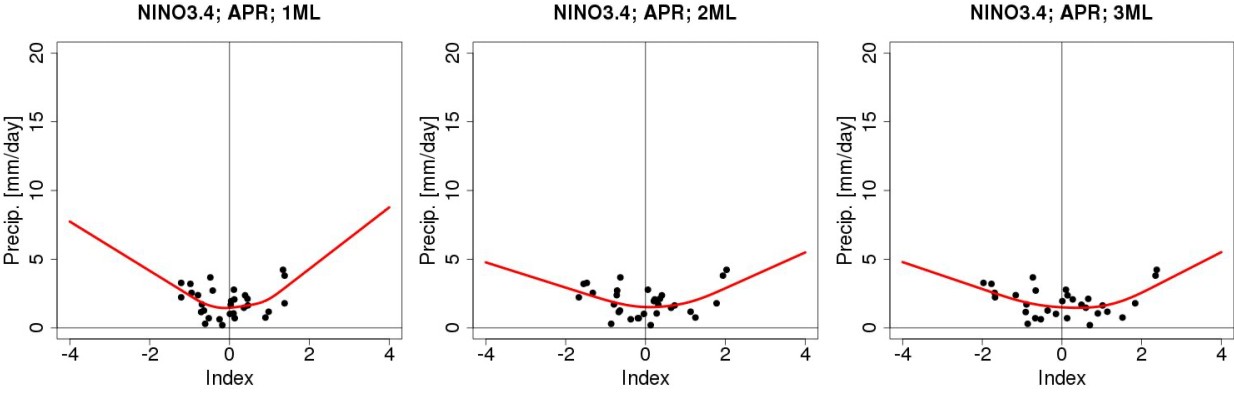





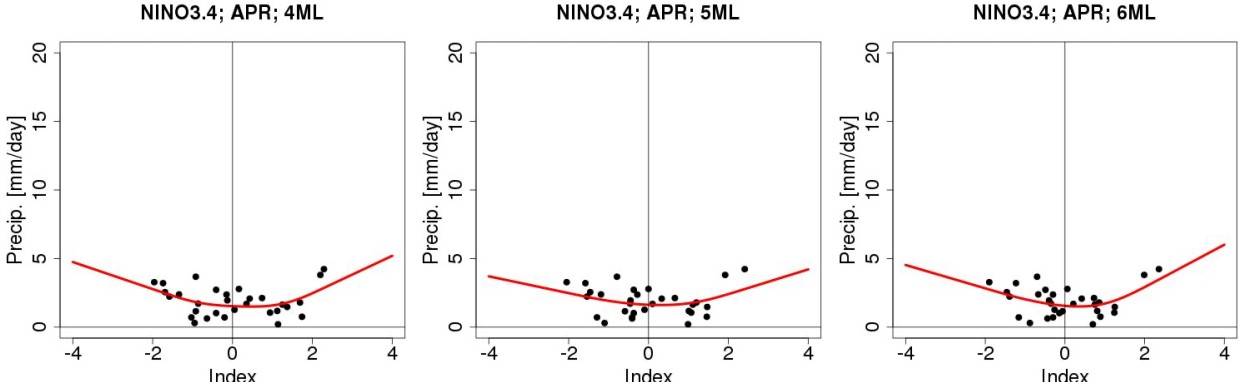

**Figure 9: Relationship between NINO3.4 and precipitation at Paris (2.5º E, 50º N). Dots indicate observational precipitation and NINO3.4 index values. Red lines are interpolated spline curves.**

495

**Table 1:  Evaluation of prediction skill in Steps 1 and 2.**

| Item | Description |
|---|---|
| **Variable** | **Precipitation** |
| **Area** | **Global** |
| **Spatial resolution** | **2.5º×2.5º (144 column; 73 rows)** |
| **Period** | **1981-2010** |
| **Time resolution** | **Monthly** |
| **Lead month of prediction** | **1-6 month** |
| **Evaluation of prediction skill** | **Anomaly correlation coefficient (ACC)** |
| **Observation** | **Global Precipitation Climatology Project (GPCP) v2.3 (regrided)** |
| **Dynamical model** | **JMA/MRI-CPS2** |
| **Statistical model** | **18 Climate indices** |

500

**Table 2: JMA/MRI-CPS2**

| | Model | Initial conditions |
|---|---|---|
| **Atmosphere** | **JMA-GSM1011C ($T_L$159L60)** | **JRA-55** |
| **Ocean** | **MRI.COM v3 (1º × 0.3-0.5º L52)** | **MOVE/MRI.COM-G2** |

**Table 3: Hindcast data of JMA/MRI-CPS2**



| Item | Description | |
|---|---|---|
| Area | Global | 405 |
| Spatial resolution | 2.5º×2.5º | |
| Period | 1979-2019 | |
| Time resolution | Daily | |
| Ensemble | 5 | 410 |
| Start day | Middle and end of month | |
| Forecast period | 240 days | |

·15   **Table 4: 18 climate indices**

| Category | Name | Long name | URL |
|---|---|---|---|
| Teleconnections | PNA | Pacific North American Index | ftp://ftp.cpc.ncep.noaa.gov/wd52dg/data/indices/pna_index.tim |
| | WP | Western Pacific Index | ftp://ftp.cpc.ncep.noaa.gov/wd52dg/data/indices/wp_index.tim |
| | EA/WR | Eastern Atlantic/Western Russia | ftp://ftp.cpc.ncep.noaa.gov/wd52dg/data/indices/eawr_index.tim |
| | NAO | North Atlantic Oscillation | ftp://ftp.cpc.ncep.noaa.gov/wd52dg/data/indices/nao_index.tim |
| | NOI | Northern Oscillation Index | https://www.pfeg.noaa.gov/products/PFEL/modeled/indices/NOIx/data/noix.txt |
| ENSO | MEI v2 | Multivariate ENSO Index | https://psl.noaa.gov/enso/mei/data/meiv2.data |
| | Nino 1+2 | Extreme Eastern Tropical Pacific SST (0-10S, 90W-80W) | http://www.cpc.ncep.noaa.gov/data/indices/ersst5.nino.mth.91-20.ascii |
| | Nino 3 | Eastern Tropical Pacific SST (5N-5S, 150W-90W) | http://www.cpc.ncep.noaa.gov/data/indices/ersst5.nino.mth.91-20.ascii |
| | Nino 4 | Central Tropical Pacific SST (5N-5S) (160E-150W) | http://www.cpc.ncep.noaa.gov/data/indices/ersst5.nino.mth.91-20.ascii |
| | Nino 3.4 | East Central Tropical Pacific SST (5N-5S) (170-120W) | http://www.cpc.ncep.noaa.gov/data/indices/ersst5.nino.mth.91-20.ascii |
| SST: Pacific (except ENSO) | WHWP | Western Hemisphere Warm Pool | https://www.esrl.noaa.gov/psd/data/correlation/whwp.data |
| | TPI(IPO) | Tripole Index for the Interdecadal Pacific Oscillation (unfiltered) | https://psl.noaa.gov/data/timeseries/IPOTPI/tpi.timeseries.ersstv5.data |
| SST: Atlantic (except WHWP) | TNA | Tropical Northern Atlantic Index | https://www.esrl.noaa.gov/psd/data/correlation/tna.data |
| | TSA | Tropical Southern Atlantic Index | https://www.esrl.noaa.gov/psd/data/correlation/tsa.data |





| | QBO | Quasi-Biennial Oscillation | https://www.esrl.noaa.gov/psd/data/correlation/qbo.data |
|---|---|---|---|
| | SOI | Southern Oscillation Index | https://www.esrl.noaa.gov/psd/data/correlation/soi.data |
| Atmosphere | AAO | Antarctic Oscillation | http://www.cpc.ncep.noaa.gov/products/precip/CWlink/daily_ao_index/aao/monthly.aao.index.b79.current.ascii |
| | AO | Antarctic Oscillation | http://www.cpc.ncep.noaa.gov/products/precip/CWlink/daily_ao_index/monthly.ao.index.b50.current.ascii |