# Peer review of "Systematic global evaluation of accuracy of seasonal climate forecasts for monthly precipitation of JMA/MRI-CPS2 by comparing with a statistical system using climate indices"

_Geoscientific Model Development, 2021_

## Referee Comment (RC1)

**Review of manuscript GMD-2021-131 entitled "Systematic global evaluation of accuracy of seasonal climate forecasts for monthly precipitation of JMA/MRI-CPS2 by comparing with a statistical system using climate indices" by Yuji Masutomi et al.**

**OVERALL RECOMMENDATION**

Reject

**SUMMARY**

This study proposes to compare the performances in monthly precipitation prediction between a newly-released dynamical seasonal forecasting system and a statistical model based on 18 climate indices. The analysis is carried out at the global scale for 12 initialization dates (at the end of each month) over a 30-year reforecast period (1981-2010). Lead months 1 to 6 are considered. The authors conclude that the dynamical system is more accurate for month-1 lead time but is superseded by the statistical model from month-2 onward (except in the 20°S-20°N equatorial region).

Note: There is trouble with the line numbering of the manuscript. The line counter is reset to 0 at section 2.2. Here, I refer to the line numbers as they appear, although they are erroneous.

**MAJOR COMMENTS**

The idea of using a statistical model as a benchmark is quite relevant for the evaluation of dynamical seasonal forecasts, and the concept of this study could have led to a valuable contribution to this field. However, the manuscript suffers from a lack of clarity and substantial flaws, so it fails to fulfill the promises it bears in the abstract. I encourage the authors to carry on this study and re-submit a new, enriched version, but I think these modifications are beyond a major revision. This why my recommendation is to reject the current manuscript.

Here are my major concerns:

1) The test that is used to define a significant ACC at the 95% level should be named and described, as it is a key component of the results.

2) The definitions of the "Ratio of sig." and "Ratio of higher ACC" indicators that are represented in Figures 5 and 8 are not clear at all, while these indicators play a central role in the interpretation of the results. Therefore, they should be explained in more details in Section 2.1 or in Section 3.1 (l. 40-47).

3) The construction of the statistical forecasts is very unclear too:

a) I do not understand why there is a separation between Step 2-1 and Step 2-2. From what I understand, leave-one-out cross-validation must be applied in the model fitting from the outset, otherwise it seems the statistical model is fit by including unknown data to be predicted.

b) I do not understand either how a single statistical forecast is obtained from the 18 statistical models (Step 2-3). I know it is the purpose of lines 84-88 (Section 2.1) to explain it, but they are actually very confusing.

4) The abstract claims that the statistical model can be used to diagnose slow dynamics that are not well reproduced by the dynamical model. This would be the most important contribution of this article, but the authors do not address it extensively while they have 18 climate indices available.

I guess Section 3.3 and Figure 9 are meant to illustrate this point, but they fail to convince. Indeed, to my understanding, the figure is purely observational and there is no analysis of the model behavior relative to the relationship between Nino 3.4 and precipitation in Paris. Then, I cannot see how it is possible to conclude "the slow dynamics (...) are not reproduced in JMA/MRI-CPS2" (l. 79-80).

5) Section 4 (Discussion) should be thoroughly re-organized, re-written and possibly merged with Section 5 (Conclusion). In its present form, I feel it only rephrases Section 3 and does not bring any additional insight.

**MINOR COMMENTS**

l. 19-22: The sentences should be switched: mention the comparison at the global scale first, before going into details about the 10°S-10°N equatorial band.

l. 30-31: "which is implemented in most SCF systems". Unnecessary, I suggest removing.

l. 37-38 vs l. 91-92:
"The mean square skill score is often used to evaluate the forecast accuracy of SCF systems" (l. 37-38)
"The anomaly correlation coefficient (ACC) between the forecast and observed values was used to evaluate the forecast accuracy" (l. 91-92).
It is strange that the MSSS is mentioned in the introduction while the whole assessment of accuracy is based on the ACC. I suggest mentioning the ACC from the outset, while removing the MSSS.

l. 34-37: "For dynamical SCF (…) with a smaller cost." This sentence is quite long and intricate, I suggest splitting and/or rephrasing for the sake of clarity.

l. 48: "**and the** Madden-Julian Oscillation"

l. 50: I would rather say "The predictability **in** St-SCFs" rather than "The predictability of St-SCFs".

l. 76-79: For the sake of clarity, I suggest trimming and rephrasing the sentences.

l. 89: "the forecast accuracy of JMA/MRI-CPS2 " Avoid repetition

l. 91: The "ACC" term is ambiguous, as it has different meanings across various studies. It might as well designate the correlation of spatial patterns or the temporal correlation between time series. From the results in the manuscript, I assume that it corresponds to a temporal correlation. Then I suggest using another expression.

l.93: "A significance level of 0.05 was used to evaluate statistical significance of ACC" Please mention the significance test here (see Major comment #1).

l. 93-94: "Forecasts with 1 to 6 lead months were evaluated". Note that there might a conflicting naming convention of lead times with other works on seasonal forecasting. For instance, in the operational Copernicus C3S seasonal forecasts (https://climate.copernicus.eu/seasonal-forecasts), if we consider forecasts initialized on September $1^{st}$, the month of September is lead time 0-month, while October is lead time 1-month. Although it is of minor importance, I am unsure what you designate by month-1, month-2, etc.

l. 109-110: "The hindcast data included five ensembles with different initial conditions (...). There were two forecasts starting in the middle and end of each month."
Something is unclear about the forecasting system setup: do you mean your ensemble forecast is a lagged-ensemble with two forecasts initialized in the middle of the month and two forecasts at the end? If so, where does the fifth member come from? And if not, do you have 5 members launched in burst mode at the end of the month (e.g September 28)?
→ Suggestion: The last two remarks (l. 93-94 and l. 109-110) could be clarified with a simple diagram for a representative start date.

l. 65, 67: "**above** 20°**N** and **below** 20°**S**"

Figure 5, caption:
"ratio of higher ACC with significant between JMA/MRI-CPS2 and St-SCFs"
I do not understand the sentence, some words must be missing or jumbled up.

---

## Author Comment (AC1)

**Responses to referees' comments**

**To referee #1**

**Major comments:**

##############################################################################

*1) The test that is used to define a significant ACC at the 95% level should be named and described, as it is a key component of the results.*

##############################################################################

We named it as CC-sig and added a description on it in the revised manuscript, as follows (L97–100):

> "The correlation coefficient (CC) between the forecast and observed values from 1981 to 2010 was used to evaluate the forecast accuracy. The statistical significance of CC was tested with a significance level of 0.05 (i.e. $p < 0.05$). The CC of each grid is shown in the map only in the grids with a significant CC. Hence, the significant CC in maps is referred to as CC-sig in the present paper."

##############################################################################

*2) The definitions of the "Ratio of sig." and "Ratio of higher ACC" indicators that are represented in Figures 5 and 8 are not clear at all, while these indicators play a central role in the interpretation of the results. Therefore, they should be explained in more details in Section 2.1 or in Section 3.1 (l. 40-47).*

##############################################################################

We added a description and a figure for explaining the indicators, as follows (L100–103 and Figure 3):

> "To evaluate the global accuracy of forecasts, we also calculated the following three types of CC: (i) CC-av: globally averaged CC with weights of grid areas; (ii) CC-rs: the

ratio of area with significant and positive CC; and (iii) CC-hi: the ratio of area with significant, positive, and higher CC between JMA/MRI-CPS2 forecasts and St-SCFs. The calculation of the three global CCs is described in Fig. 3."

[Figure]

**Figure 3: Example of calculation of (i) CC-av: CC globally averaged with weights of grid areas; (ii) CC-rs: the ratio of area with significant and positive CC; and (iii) CC-hi: the ratio of area with significant, positive, and higher CC between JMA/MRI-CPS2 forecasts and St-SCFs. The boxes represent grids and the numbers in the boxes indicate the grid area and CC. The symbol * shows that the CC is significant.**

##################################################################################

*3) The construction of the statistical forecasts is very unclear too:*

*a) I do not understand why there is a separation between Step 2-1 and Step 2-2. From what I understand, leave-one-out cross-validation must be applied in the model fitting from the outset, otherwise it seems the statistical model is fit by including unknown data to be predicted.*

*b) I do not understand either how a single statistical forecast is obtained from the 18 statistical models (Step 2-3). I know it is the purpose of lines 84-88 (Section 2.1) to explain it, but they are actually very confusing.*

##############################################################################

We agree with your comments that the description of the statistical models was confusing in the previous manuscript. We combined Steps 2-1 and 2-2 and modified the description and Figure 1. We guess that the unclearness regarding the point (a) caused the confusion of the point (b). Although we did not modified the description on the selection of the statistical models, we believe it became clear. The modified description and figure are as follows (L81–85 and Figure 1):

"Next, in Step 2-1, 18 monthly precipitation forecasts were produced for each grid by using 18 climate indices and GPCP v2.3. The following leave-one-out method was used for producing the forecasts. First, the data of the forecast year was removed before constructing statistical models with monthly values of a climate index as the explanatory variable and monthly precipitation as the objective variable (see Section 2.2.2) through the smoothing spline method. Then, the forecast values were obtained using the constructed statistical models and removed data as inputs for the models."

[Figure]

**Figure 1: Research outline**

##################################################################################

*4) The abstract claims that the statistical model can be used to diagnose slow dynamics that are not well reproduced by the dynamical model. This would be the most important contribution of this article, but the authors do not address it extensively while they have 18 climate indices available. I guess Section 3.3 and Figure 9 are meant to illustrate this point, but they fail to convince. Indeed, to my understanding, the figure is purely observational and there is no analysis of the model behavior relative to the relationship between Nino 3.4 and precipitation in Paris. Then, I cannot see how it is possible to conclude "the slow dynamics (...) are not reproduced in JMA/MRI-CPS2" (l. 79-80).*

##################################################################################

We added a description and figure showing the comparison of precipitation in Paris from 1981-2010 between observations and forecasts in JMA/MRI-CPS2 and St-SCF using NINO3 (we used NINO3 in the revised manuscript instead of NINO3.4) so that the difference between JMA/MRI-CPS2 and St-SCF using NINO3 became clear (L185–190 and Figure 12).

"Fig. 12 shows the comparison of precipitation in Paris (2.5° E and 50º N) from 1981–2010 between observations and forecasts for zero to two months in advance by JMA/MRI-CPS2 and St-SCFs using the NINO3 index. The CC and significance are also shown in Fig. 12. The CCs of St-SCFs using the NINO3 index were significant and higher than those in JMA/MRI-CPS2. In particular, the St-SCFs using NINO3 index could forecast the highest precipitation of 1983 and 1998 among the 30 years even for forecasts two months in advance, while JMA/MRI-CPS2 could only forecast the high precipitation of 1981 for forecasts zero month in advance and could not forecast the others."

[Figure]

[Figure]

**Figure 12: Comparison of precipitation at Paris (2.5º E, 50º N) from 1981–2010 between observations (GPCP: black circle) and forecasts (red dots) by JMA/MRI-CPS2 and St-SCFs with NINO3. Correlation coefficient (CC) and statistical significance (***: p<0.001; **: p<0.01; *: p<0.05) are also shown.**

################################################################################

*5) Section 4 (Discussion) should be thoroughly re-organized, re-written and possibly merged with Section 5 (Conclusion). In its present form, I feel it only rephrases Section 3 and does not bring any additional insight.*

################################################################################

We drastically simplified Section 4 (Discussion) to make the findings clear and merged it with Section 5 (Conclusion).

**Minor comments:**

################################################################################

*l. 19-22: The sentences should be switched: mention the comparison at the global scale first, before going into details about the 10°S-10°N equatorial band.*

################################################################################

We switched the sentences in the revised manuscript (L19–21).

############################################################################

*l. 30-31: "which is implemented in most SCF systems". Unnecessary, I suggest removing.*

############################################################################

We removed it (L30).

############################################################################

*l. 37-38 vs l. 91-92:*

*"The mean square skill score is often used to evaluate the forecast accuracy of SCF systems" (l. 37-38)*

*"The anomaly correlation coefficient (ACC) between the forecast and observed values was used to evaluate the forecast accuracy" (l. 91-92).*

*It is strange that the MSSS is mentioned in the introduction while the whole assessment of accuracy is based on the ACC. I suggest mentioning the ACC from the outset, while removing the MSS*

############################################################################

We removed the description on the MSSS (L35).

############################################################################

*l. 34-37: "For dynamical SCF (...) with a smaller cost." This sentence is quite long and intricate, I suggest splitting and/or rephrasing for the sake of clarity.*

############################################################################

We simplified the sentence by removing some words, as follows (L33–35):

> "For dynamical SCF (Dyn-SCF) systems, in particular those with a large computational load, it is necessary to show the added values as benefits to weigh out the large cost by comparing them with the forecast accuracy of simple methods with a smaller cost."

######################################################################

*l. 48: "**and the** Madden-Julian Oscillation"*

######################################################################

We modified it (L44).

######################################################################

*l. 50: I would rather say "The predictability **in** St-SCFs" rather than "The predictability of St-SCFs".*

######################################################################

We modified it (L46).

######################################################################

*l. 76-79: For the sake of clarity, I suggest trimming and rephrasing the sentences.*

######################################################################

We modified them, as follows (L76–79):

> "The evaluation was conducted globally at a spatial resolution of 2.5° × 2.5°. Before the evaluation, GPCP v2.3 was re-gridded to the grid of JMR/MRI-CPS2 using the bilinear method, because the center of their grids did not match even though the spatial resolution of both JMA/MRI-CPS2 and GPCP v2.3 was 2.5° × 2.5°."

######################################################################

*l. 89: "the forecast accuracy of JMA/MRI-CPS2 forecasts" Avoid repetition*

######################################################################

We removed the word (L90).

###############################################################################

*l. 91: The "ACC" term is ambiguous, as it has different meanings across various studies. It might as well designate the correlation of spatial patterns or the temporal correlation between time series. From the results in the manuscript, I assume that it corresponds to a temporal correlation. Then I suggest using another expression.*

###############################################################################

We used "CC" instead of "ACC" in the revised manuscript and simplified the explanation on coefficient correlations, as follows (L97–98):

> "The correlation coefficient (CC) between the forecast and observed values from 1981 to 2010 was used to evaluate the forecast accuracy."

###############################################################################

*l.93: "A significance level of 0.05 was used to evaluate statistical significance of ACC" Please mention the significance test here (see Major comment #1).*

###############################################################################

We modified the sentence as follows (L98):

> "The statistical significance of CC was tested with a significance level of 0.05 (i.e. $p < 0.05$)."

###############################################################################

*l. 93-94: "Forecasts with 1 to 6 lead months were evaluated". Note that there might a conflicting naming convention of lead times with other works on seasonal forecasting. For instance, in the operational Copernicus C3S seasonal forecasts (https://climate.copernicus.eu/seasonal-forecasts), if we consider forecasts initialized on*

*September 1st, the month of September is lead time 0-month, while October is lead time 1-month. Although it is of minor importance, I am unsure what you designate by month-1, month-2, etc.*

\#\#\#\#\#\#\#\#\#\#\#\#\#\#\#\#\#\#\#\#\#\#\#\#\#\#\#\#\#\#\#\#\#\#\#\#\#\#\#\#\#\#\#\#\#\#\#\#\#\#\#\#\#\#\#\#\#\#\#\#\#\#\#\#\#\#\#\#\#\#\#\#\#\#\#\#\#\#\#\#\#

We changed the counting method of lead months so that lead months in the present study are similar to those in the C3S systems. In addition, we modified the sentences for the explanation on lead months and added a figure to explain lead months for forecasts, as follows (L92–96 and Figure 2):

"Forecasts made 0 to 5 months in advance were evaluated. Fig. 2 shows an example of 5-month lead forecasts in JMA/MRI-CPS2 and St-SCFs using climate indices. For example, in the case of precipitation forecasts for July, monthly precipitation forecasts that started from January were used for 5-month lead forecasts. In the statistical model for forecasts for precipitation in July, the precipitation in July was used as an objective variable and climate indices in January were used as explanatory variables."

**Example of 5-month lead forecast**

**Figure 2: Lead month for forecasts in JMA/MRI-CPS2 and St-SCFs using climate indices (example of 5-month lead forecasts)**

\#\#\#\#\#\#\#\#\#\#\#\#\#\#\#\#\#\#\#\#\#\#\#\#\#\#\#\#\#\#\#\#\#\#\#\#\#\#\#\#\#\#\#\#\#\#\#\#\#\#\#\#\#\#\#\#\#\#\#\#\#\#\#\#\#\#\#\#\#\#\#\#\#\#\#\#\#\#\#\#\#

*l. 109-110: "The hindcast data included five ensembles with different initial conditions (...).*
*There were two forecasts starting in the middle and end of each month."*
*Something is unclear about the forecasting system setup: do you mean your ensemble forecast*
*is a lagged-ensemble with two forecasts initialized in the middle of the month and two forecasts*
*at the end? If so, where does the fifth member come from? And if not, do you have 5 members*
*launched in burst mode at the end of the month (e.g September 28)?*
*→ Suggestion: The last two remarks (l. 93-94 and l. 109-110) could be clarified with a simple*
*diagram for a representative start date.*

###############################################################################

We added a figure for explaining the forecast system in JMA/MRI-CPS2 and lead months
(Figure 2)

###############################################################################

*l. 65, 67: "**above** 20°N and **below** 20°**S**"*

###############################################################################

We modified the words (L176–177).

###############################################################################

*Figure 5, caption:*
*"ratio of higher ACC with significant between JMA/MRI-CPS2 and St-SCFs" I do not*
*understand the sentence, some words must be missing or jumbled up.*

###############################################################################

We added a figure explaining the indicators (Figure 3).

---

## Author Comment (AC3)

**Responses to referees' comments**

**To referee #2**

**Major comments:**

The description on the development of statistical models was confusing in the previous manuscript. Actually we first constructed statistical models by removing the data in forecast years and then produced precipitation forecasts by using the removed data. We improved the explanation on the development of the statistical models and modified Figure 1, as follows:

"Next, in Step 2-1, 18 monthly precipitation forecasts were produced for each grid by using 18 climate indices and GPCP v2.3. The following leave-one-out method was used for producing the forecasts. First, the data of the forecast year was removed before constructing statistical models with monthly values of a climate index as the explanatory variable and monthly precipitation as the objective variable (see Section 2.2.2) through the smoothing spline method. Then, the forecast values were obtained using the constructed statistical models and removed data as inputs for the models."

Figure 1: Research outline

Thank you for your very important comment. We recognize that it is a very important issue to compare the accuracy of forecasts, taking into account the uncertainty of forecasts. However, this includes a difficult challenge as described below. Therefore, we would like to make this a future challenge and describe it in Section 4 (Discussion and conclusions).

First of all, even in statistical models, it is possible to show the uncertainty of forecasts by showing the confidence interval of forecasts or to make ensembles by using resampling methods such as the bootstrap method. Moreover, by using the resampling methods, statistical forecast systems can produce very large ensembles (e.g., 10,000) much more easily than dynamical ones. In order to compare the accuracy of two systems that handle uncertainty in a different way, we must first

understand the characteristics of the uncertainty in each system, and then consider how to compare them. For example, how do we compare the accuracy of a Dyn-SCF with an ensemble of approximately 10 and a St-SCF with an ensemble of about 10,000, taking uncertainty into account? We believe that intensive studies on this issue are needed. However, this is clearly beyond the scope of the present research. We would like to discuss the point in Section 4 (Discussion and conclusions) as follows:

"How to compare the accuracy of Dyn-SCFs and St-SCFs considering the uncertainty in forecasts is an important future challenge. Forecasting inherently includes uncertainty. For this reason, ensemble forecasting is widely conducted in Dyn-SCFs systems. In St-SCFs systems, a huge ensemble of forecasts can be easily made by using resampling methods such as the bootstrap method or the confidence interval of the forecasts can be calculated. In order to compare the accuracy of two forecast systems that handle uncertainty in a different way, we must first understand the characteristics of the uncertainty in each system, and then consider how to compare them. For example, how do we compare the accuracy of a Dyn-SCF with an ensemble of approximately 10 and a St-SCF with an ensemble of about 10,000, taking uncertainty into account? We believe that intensive studies on this issue are needed and it is expected that such studies will be conducted in the future."

**Minor comments:**

"Precipitation forecasting is very important for effective water management and disaster reduction. It has been shown that the accuracy of precipitation forecasts in Dyn-SCF systems is lower than that that of temperature forecasts, and areas with highly accurate precipitation forecasts is limited in the tropics (Doblas-Reyes et al., 2013). So far, it is

not well understood to what extent the accuracy of Dyn-SCF systems for precipitation forecasts adds value compared to St-SCF systems."

In addition, we added the explanation that statistical forecast systems can be alternative for dynamical ones in case of focusing a small number of climate variables, as follows:

"In case of forecasting a small number of specific climate variables, statistical SCF (St-SCF) systems are an alternative and simpler method for Dyn-SCF systems"

In the revised manuscript, we added a figure showing the selected climate indices for each grid. This figure shows that the same index tends to be selected in each small area. This implies that St-SCFs using climate indices have a consistent spatial forecast within each small area. We added the following figure and the explanation above, as follows: